# Different renoprotective effects of luseogliflozin depend on the renal function at the baseline in patients with type 2 diabetes: A retrospective study during 12 months before and after initiation

**Hiroyuki Ito**[ORCID]*, **Suzuko Matsumoto, Takuma Izutsu, Eiji Kusano, Jiro Kondo, Hideyuki Inoue, Shinichi Antoku, Tomoko Yamasaki, Toshiko Mori, Michiko Togane**

Department of Diabetes, Metabolism and Kidney Disease, Edogawa Hospital, Edogawa, Tokyo, Japan

* ito@edogawa.or.jp

## Abstract

### Aims

The safety and efficacy, particularly, the factors associated with the renal prognosis, were assessed over 12 months after the initiation of luseogliflozin therapy in Japanese patients with type 2 diabetes and renal impairment.

### Methods

In total, 238 patients treated with luseogliflozin (2.5 mg, once daily) were studied as the safety analysis set. Two hundred and two subjects whose medication was continued over 12 months were investigated as the full analysis set. The subjects were divided into 3 groups based on the estimated glomerular filtration rate (eGFR): high eGFR ($n = 49$), normal eGFR ($n = 116$) and low eGFR ($n = 37$) groups.

### Results

The body weight, systolic blood pressure, HbA1c and urinary protein excretion gradually decreased from baseline in all eGFR groups. While the eGFR was significantly reduced from baseline in the high and normal eGFR groups, the eGFR did not significantly differ over time in the low eGFR group. There was no marked difference in the frequency of adverse events that were specific for SGLT2 inhibitors among the 3 groups in the safety analysis set.

### Conclusions

Luseogliflozin can preserve the renal function in the medium term in patients with type 2 diabetes and renal impairment without an increase in specific adverse events.

**Data Availability Statement:** All relevant data are within the manuscript and its Supporting Information files.

**Funding:** This work was partly supported by Taisho Pharmaceutical Co., Ltd (https://www.taisho.co.jp/). The funders had no role in study design, data collection and analysis, decision to publish, or preparation of the manuscript. There was no additional external funding received for this study.

**Competing interests:** H Ito has received funding support and lecture fees from Taisho Pharmaceutical Co., Ltd., and lecture fees from Eli Lilly Japan KK, Boehringer Ingelheim, Takeda Pharmaceutical Company Ltd., Sanofi KK, Novo Nordisk Pharma Ltd., MSD KK, Novartis Pharma KK, Astellas Pharma, Daiichi Sankyo Company, Terumo Corporation, Mochida Pharmaceuticals, Teijin Pharma, Kissei Pharmaceuticals, Kowa Pharmaceuticals, Mitsubishi Tanabe Pharma Corporation, Sanwa Kagaku Kenkyusho, Dainippon Sumitomo Pharma, AstraZeneca KK, Kyowa Hakko Kirin, Shionogi and Co, Otsuka Pharmaceutical Co., Ltd., Bayer Yakuhin, Ltd., and Santen Pharmaceutical Co., Ltd., and has received consulting fee from Becton, Dickinson and Company. S Matsumoto has received lecture fees from Novo Nordisk Pharma Ltd., Astellas Pharma, and AstraZeneca KK. T Izutsu has received lecture fees from Sanofi KK, Taisho Pharmaceutical Co., Ltd., Kyowa Hakko Kirin, Bayer Yakuhin, Ltd., and Mitsubishi Tanabe Pharma Corporation. S Antoku has received lecture fees from Kyowa Hakko Kirin, Sanofi KK, Kyowa Hakko Kirin, Taisho Pharmaceutical Co., Ltd., Daiichi Sankyo Company, and Otsuka Pharmaceutical Co., Ltd. E Kusano, J Kondo, H Inoue, T Yamasaki, T Mori and M Togane have no conflict of interest. This does not alter our adherence to PLOS ONE policies on sharing data and materials.

## Introduction

Sodium-glucose cotransporter 2 (SGLT2) inhibitors was not previously recommended for use in patients with type 2 diabetes and renal impairment because their blood glucose lowering effect was insufficient [1]. However, several clinical studies performed in patients with type 2 diabetes have shown that SGLT2 inhibitors suppress the development of diabetic macroangiopathies and heart failure and also act as renal protectors [2–7], including in Asian subjects, whose body mass index (BMI) and estimated glomerular filtration rate (eGFR) are generally lower than in Western populations [8, 9].

When SGLT2 inhibitors were introduced to Japan, where elderly patients with type 2 diabetes account for much of the population [10, 11], there were concerns about drug-related adverse events (AEs), such as fluid depletion and cerebral infarction. Therefore, the prescription rate of SGLT2 inhibitors had not been high [12], with dipeptidyl peptidase-4 (DPP-4) inhibitors most commonly used in real-world clinical practice for the treatment of Japanese patients with type 2 diabetes [13, 14] because of their safety and reliability in glycemic control among elderly patients and subjects with renal impairment [15, 16]. However, there has been no clear evidence to support the inhibition of cardiovascular events or diabetic nephropathy by DPP-4 inhibitors [17–20], although the glucose-lowering efficacy of DPP-4 inhibitors in patients with type 2 diabetes is better in Asians than in other ethnic groups [21].

Recently, it was reported that ipragliflozin therapy was well tolerated and reduced surrogate endpoints of diabetic vascular complications, such as HbA1c, body weight and blood pressure, in elderly Japanese patients with type 2 diabetes based on a post-marketing surveillance study [22, 23]. Furthermore, we reported the safety and efficacy of empagliflozin through the improvement of the blood glucose level, body weight, liver function, hemoglobin value and serum high-density lipoprotein (HDL)-cholesterol concentration in elderly Japanese patients with type 2 diabetes whose renal function was lower than that in non-elderly patients [24].

Cardiovascular events and end-stage kidney disease occur more frequently in patients with type 2 diabetes and renal impairment than in those with a normal kidney function [25], including in Japanese populations [26–30]. Therefore, the safety and efficacy of SGLT2 inhibitors should be determined in Japanese patients with type 2 diabetes and renal impairment. It was previously reported that the blood glucose-lowering effect of the SGLT2 inhibitor luseogliflozin is worse in type 2 diabetic patients with renal impairment than in those with a normal renal function [31–33]. Although the initial reduction in the eGFR after luseogliflozin administration seems smaller in patients with type 2 diabetes and renal impairment (eGFR 30 to <60 mL/min/1.73 m$^2$) than in those with a high eGFR (eGFR ≥90 mL/min/1.73 m$^2$) based on phase III studies [31], its renoprotective effect has not been determined in real-world clinical settings.

In the present study, we retrospectively examined the safety and efficacy of luseogliflozin while paying particular attention to the factors associated with the renal prognosis over 12 months after the initiation of luseogliflozin therapy in Japanese patients with type 2 diabetes and renal impairment.

## Materials and methods

### Study design and patients

A flow chart of the patient selection process is shown in **Fig 1**. Three hundred and thirty-seven Japanese patients with type 2 diabetes who received 2.5 mg of luseogliflozin once daily (Lusefi® tablets; Taisho Pharmaceutical Co., Ltd., Tokyo, Japan) at our department from December 2014 to September 2018 were eligible for inclusion in this study. Subjects who had

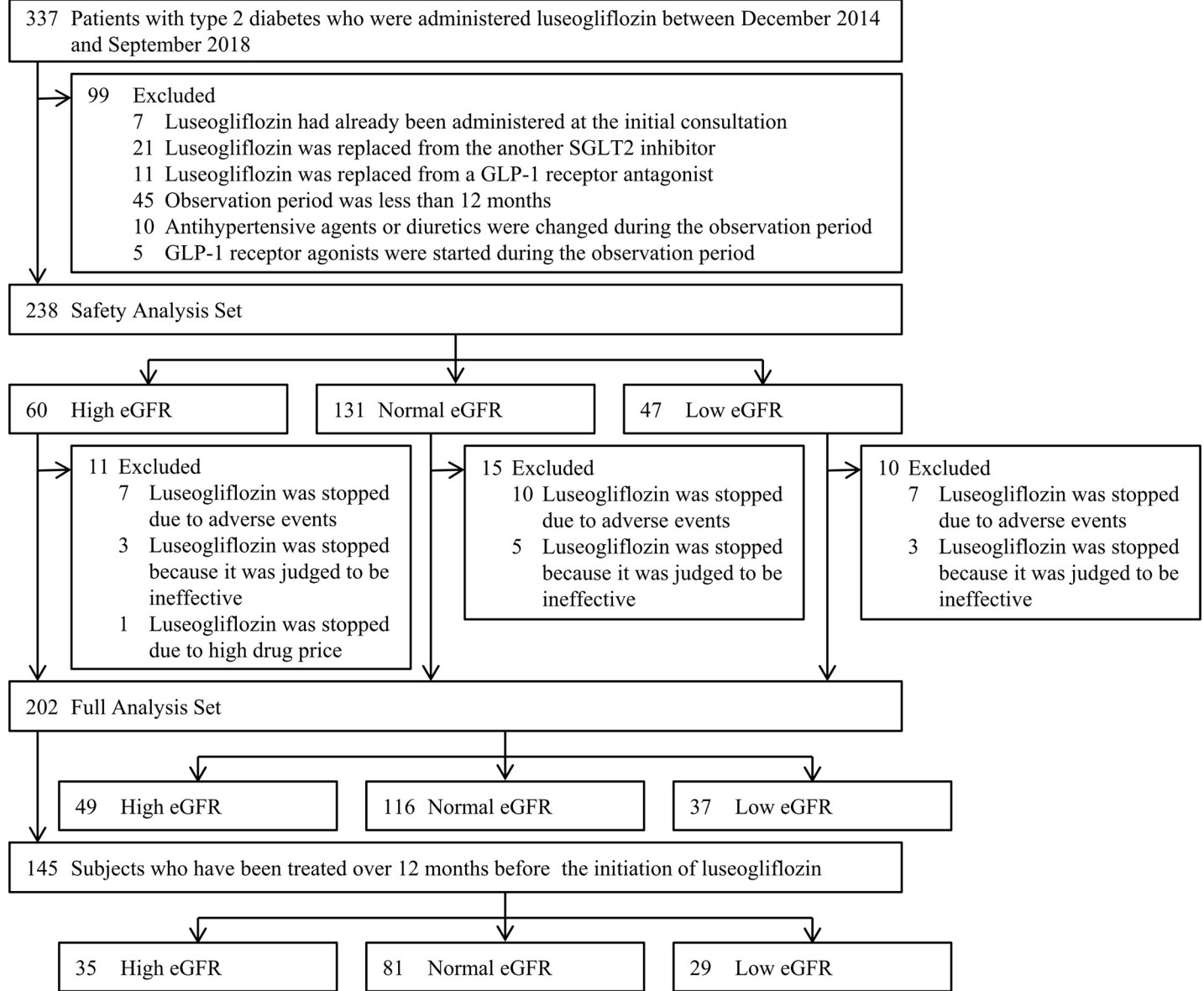

**Fig 1. Flowchart of patient selection.** The safety of luseogliflozin was analyzed in the safety analysis set ($n$ = 238) and the effectiveness was investigated in the full analysis set ($n$ = 202). The analysis sets were divided to high ($\geq$90 mL/min/1.73 m$^2$), normal (60 to <90 mL/min/1.73 m$^2$) and low (<60 mL/min/1.73 m$^2$) eGFR groups. SGLT2, Sodium-glucose cotransporter 2; GLP-1, glucagon-like peptide-1; eGFR, estimated glomerular filtration rate.

already been prescribed luseogliflozin at the initial consultation ($n$ = 7), subjects in whom luseogliflozin treatment started with the replacement of another SGLT2 inhibitor ($n$ = 21) or a glucagon-like peptide-1 (GLP-1) receptor agonist ($n$ = 11), subjects who discontinued treatment or who were transferred to other hospitals during the observation period ($n$ = 45), subjects whose antihypertensive agents were changed during the observation period ($n$ = 10), and subjects in whom GLP-1 receptor agonist treatment started during the observation period ($n$ = 5) were excluded from the analysis.

In total, 238 patients with type 2 diabetes (males: 71%, 59±12 years old) were studied as the safety analysis set (SAS) in order to analyze the safety of luseogliflozin. The SAS included 60 subjects with a high eGFR ($\geq$90 mL/min/1.73 m$^2$), 131 with a normal eGFR (60 to <90 mL/

min/1.73 m$^2$) and 47 subjects with a low eGFR (<60 mL/min/1.73 m$^2$). After excluding subjects who discontinued luseogliflozin treatment due to any AEs ($n$ = 24), a judgement of inefficacy by physicians ($n$ = 11) or complaints of high drug price by the patient ($n$ = 1), 202 subjects were investigated as the full analysis set (FAS) in order to assess the effectiveness of luseogliflozin. Finally, the study subjects were divided into 3 groups based on the eGFR: high eGFR ($n$ = 49), normal eGFR ($n$ = 116) and low eGFR ($n$ = 37) groups. Furthermore, the changes in the eGFR were determined in 145 subjects who had been treated in our department over 12 months before the initiation of luseogliflozin.

## Measurements

The eGFR was calculated using the formula recommended by the Japanese Society of Nephrology [34]. Diabetic nephropathy was defined as a urinary albumin-to-creatinine ratio (uACR) of ≥30 mg/g/creatinine in a random spot urine test. The urinary protein excretion (uPE) was evaluated by the pyrogallol red method using urine test-strips (Uriflet S; ARKRAY, Inc., Kyoto, Japan) and an automatic analyzer (Austin MAX AX 4280; ARKRAY, Inc.). Proteinuria was graded as (±), (1+), (2+) and (3+) corresponding to 15 mg/dL, 50 mg/dL, 150 mg/dL and 325 mg/dL, respectively, according to the median value of the measurement range in the semi-quantitative results [24, 35, 36]. Urinary liver-type fatty-acid binding protein (L-FABP), a biomarker of tubulointerstitial injury of the kidneys that predicts renal dysfunction associated with diabetic nephropathy [37–39], was measured using a chemiluminescent enzyme immunoassay at an external laboratory (SRL Co., Tokyo, Japan).

Obese individuals were defined as those with a BMI of ≥25.0 kg/m$^2$. Hypertension was defined as a systolic blood pressure of ≥140 mmHg and/or a diastolic blood pressure of ≥90 mmHg. Participants currently using antihypertensive medications were also classified as being positive for hypertension. Hyper-low-density lipoprotein (LDL)-cholesterolemia was defined as a serum LDL-cholesterol concentration of ≥ 3.62 mmol/L (140 mg/dL) or the current use of statins or ezetimibe. Hypo-HDL-cholesterolemia was defined as a serum HDL-cholesterol concentration of <1.03 mmol/L (40 mg/dL). Hyperuricemia was defined by serum uric acid levels >327 μmol/L (7.0 mg/dL) or as patients using benzbromarone, allopurinol, febuxostat, or topiroxostat. A current drinker was defined as a person consuming >20 g ethanol equivalent/day. Diabetic retinopathy was graded as simple, preproliferative, or proliferative retinopathy based on the results of a funduscopic examination performed by expert ophthalmologists. Diabetic peripheral neuropathy was diagnosed by the presence of two or more components among clinical symptoms (bilateral spontaneous pain, hypoesthesia, or paraesthesia of the legs), the absence of ankle tendon reflexes, and decreased vibration sensations using a C128 tuning fork. Cerebrovascular disease was diagnosed by the physicians as a history of an ischemic stroke using brain computed tomography or magnetic resonance imaging. Only the patients with symptoms were classified as having cerebrovascular disease, and cases of silent brain infarction, transient ischemic attack, and a brain hemorrhage were excluded from this study. Coronary heart disease was diagnosed based on a previous history of myocardial infarction, angina pectoris, electrocardiogram abnormalities suggesting myocardial ischemia, or interventions after a coronary angiographic examination. Peripheral arterial disease was diagnosed by the absence of a pulse in the legs, along with ischemic symptoms, obstructive findings on an ultrasonographic or angiographic examination of the lower extremities, or an ankle brachial pressure index <0.9.

The clinical parameters and AEs were retrospectively examined over 12 months after the initiation of luseogliflozin based on the subjects' medical records. When clinical data including the body weight, blood pressure, uPE, HbA1c and serum creatinine concentration were

missing, the appropriate value obtained on the previous visit was used according to the last observational carried forward (LOCF) method.

## Ethics conduct

The study was conducted in accordance with the principles expressed in the 2008 Declaration of Helsinki. The Ethics Committee of Edogawa Hospital approved the study protocol and waived the need for written informed consent because the data were analyzed anonymously for this retrospective analysis based on information stored in the hospital (approved number: 2018–30, approval date: October 18, 2018). The trial is registered on UMIN-CTR, identifier UMIN000041193.

## Statistical analyses

All data are presented as the mean±standard deviation. The $\chi^2$ test was used for between-group comparisons of categorical variables. None of the continuous variables (age, duration of diabetes, body weight, BMI, blood pressure, uACR, uPE, urinary L-FABP, HbA1c, serum lipid concentrations, aspartate transaminase [AST], alanine transaminase [ALT], γ-glutamyl transpeptidase [γ-GTP], creatinine, uric acid, and eGFR) showed a normal distribution in the Shapiro-Wilk tests, so the Kruskal-Wallis test and Wilcoxon's signed rank test were used to assess the significance of differences in the continuous variables. Wilcoxon's rank sum test was used to assess the significance of differences in the body weight, blood pressure, uPE, uACR, urinary L-FABP, HbA1c, and eGFR during the observation period compared to baseline values. A least squares model was used to evaluate the associations between the clinical background factors of the patients and the changes at 12 months after the initiation of luseogliflozin therapy in the eGFR and uPE. Factors that showed a significant association with changes in each dependent variable in a univariate analysis were included in a multivariate analysis. *P* values of <0.05 (two-tailed) were considered to indicate statistical significance. The statistical software package JMP version 12.2.0 (SAS Institute, Cary, NC) was used to perform all analyses. All data are present in **S1 Dataset**.

## Results

### Baseline characteristics and efficacy

The clinical characteristics of the FAS at the baseline are shown in **Table 1**. The patients were significantly older and the duration of diabetes significantly longer in the low eGFR group than in the high or normal eGFR group. The rates of diabetic nephropathy, peripheral neuropathy, hyper-LDL-cholesterolemia and hyperuricemia were significantly higher in the low eGFR group than in the high or normal eGFR group. Although the use of antihypertensive agents and the values of uACR, uPE and urinary L-FABP increased with the progression of renal impairment, there were no significant differences among the three groups. The HbA1c also did not differ among the three groups.

 **Table 2** shows the changes in clinical parameters of the FAS from baseline to 12 months after the initiation of luseogliflozin. The body weight and systolic blood pressure gradually decreased from the baseline value in all eGFR groups while the diastolic blood pressure did not significantly differ among groups. The changes in body weight and blood pressure were similar in the three groups. The HbA1c was also reduced in all eGFR groups after luseogliflozin administration, and the change in HbA1c tended to be smaller in the group with progressive renal impairment. The change in HbA1c from baseline was significantly smaller in the low eGFR group than in the high eGFR group (*P* = 0.01). The uPE, uACR and urinary L-FABP values significantly decreased in all eGFR groups. The change in the uPE from baseline was

**Table 1. The clinical characteristics of the full analysis set at baseline.**

| | N¶ | All subjects | Groups according to eGFR at baseline | | | P |
|---|---|---|---|---|---|---|
| | | | High eGFR | Normal eGFR | Low eGFR | |
| | | | (*n* = 49) | (*n* = 116) | (*n* = 37) | |
| Male (%) | 202 | 71 | 69 | 72 | 70 | 0.92 |
| Age (years) | 202 | 59±11 | 53±11 | 58±10** | 67±10**, ## | <0.01 |
| Duration of diabetes (years) | 187 | 10±8 | 8±6 | 9±7 | 13±9* | 0.04 |
| Smoking history (%) | 178 | 60 | 68 | 57 | 56 | 0.40 |
| Current drinker (%) | 192 | 26 | 40 | 23 | 17 | 0.04 |
| Body weight (kg) | 173 | 77.2±17.9 | 77.5±20.6 | 76.8±17.2 | 77.9±16.7 | 0.99 |
| Body mass index (kg/m$^2$) | 173 | 28.5±5.6 | 28.5±5.6 | 28.1±5.4 | 29.7±6.3 | 0.44 |
| Systolic blood pressure (mmHg) | 193 | 134±15 | 137±16 | 134±15 | 132±14 | 0.64 |
| Diastolic blood pressure (mmHg) | 193 | 81±12 | 84±14 | 81±11 | 75±11**, ## | <0.01 |
| Diabetic retinopathy (%)† | 175 | 47 | 42 | 48 | 52 | 0.69 |
| Diabetic nephropathy (%) | 196 | 50 | 50 | 44 | 69 | 0.03 |
| Diabetic peripheral neuropathy (%) | 117 | 45 | 38 | 39 | 45 | 0.02 |
| Cerebrovascular disease (%) | 202 | 6 | 4 | 5 | 11 | 0.37 |
| Coronary heart disease (%) | 202 | 14 | 6 | 14 | 24 | 0.05 |
| Peripheral arterial disease (%) | 202 | 2 | 0 | 2 | 5 | 0.20 |
| Obesity (%) | 173 | 76 | 70 | 74 | 91 | 0.10 |
| Hypertension (%) | 202 | 76 | 73 | 74 | 84 | 0.45 |
| Hyper-LDL-cholesterolemia (%) | 202 | 82 | 71 | 84 | 92 | 0.04 |
| Hypo-HDL-cholesterolemia (%) | 182 | 27 | 24 | 27 | 31 | 0.75 |
| Hyperuricemia (%) | 202 | 17 | 8 | 14 | 38 | <0.01 |
| RAAS inhibitors use (%)‡ | 202 | 55 | 47 | 53 | 70 | 0.09 |
| Calcium channel blockers use (%) | 202 | 43 | 37 | 41 | 57 | 0.14 |
| Cholesterol lowering agents use (%) | 202 | 74 | 61 | 76 | 86 | 0.03 |
| Urate lowering agents use (%) | 202 | 13 | 8 | 9 | 35 | <0.01 |
| Anti-diabetic agents use (%) | 202 | | | | | |
| Metformin | | 66 | 67 | 69 | 54 | 0.24 |
| Sulfonylureas | | 19 | 20 | 20 | 16 | 0.87 |
| Thiazolidinediones | | 12 | 16 | 12 | 8 | 0.51 |
| α-glucosidase inhibitors | | 12 | 10 | 11 | 16 | 0.66 |
| Glinides | | 5 | 4 | 5 | 5 | 0.95 |
| DPP-4 inhibitors | | 66 | 61 | 68 | 68 | 0.68 |
| GLP-1 receptor agonists | | 4 | 0 | 3 | 11 | 0.04 |
| Insulin | | 27 | 20 | 26 | 38 | 0.19 |
| Number of anti-diabetic agents | | 2.1±1.1 | 2.0±1.1 | 2.1±1.1 | 2.2±1.0 | 0.24 |
| uPE (mg/dL) | 196 | 22±47 | 13±19 | 23±50 | 33±58 | 0.10 |
| uACR (mg/gCr) | 157 | 117±274 | 91±162 | 112±249 | 178±449 | 0.34 |
| Urinary L-FABP (μg/gCr) | 50 | 5.2±6.9 | 3.6±2.0 | 3.6±3.1 | 9.2±11.7 | 0.11 |
| HbA1c (%) | 201 | 8.3±1.4 | 8.5±1.6 | 8.3±1.4 | 8.1±1.1 | 0.61 |
| HbA1c (mmol/mol) | 201 | 67±15 | 69±17 | 67±15 | 65±12 | 0.61 |
| LDL-cholesterol (mmol/L) | 183 | 2.79±0.85 | 2.99±0.86 | 2.77±0.81 | 2.60±0.91** | 0.04 |
| HDL-cholesterol (mmol/L) | 182 | 1.25±0.33 | 1.24±0.33 | 1.28±0.34 | 1.18±0.29 | 0.31 |
| AST (IU/L) | 199 | 30±21 | 31±24 | 29±16 | 32±29 | 0.83 |
| ALT (IU/L) | 199 | 35±31 | 39±35 | 35±30 | 32±27 | 0.49 |
| γGTP (IU/L) | 189 | 59±58 | 77±73 | 56±56 | 45±34**, # | 0.02 |
| Creatinine (μmol/L) | 202 | 70.6±22.3 | 51.8±8.4 | 68.5±11.5** | 101.9±27.5**, ## | <0.01 |

*(Continued)*

**Table 1.** (Continued)

| | N¶ | All subjects | Groups according to eGFR at baseline | | | P |
| | | | High eGFR | Normal eGFR | Low eGFR | |
| | | | (*n* = 49) | (*n* = 116) | (*n* = 37) | |
| Uric acid (μmol/L) | 137 | 322±68 | 288±63 | 324±70* | 353±49**, # | <0.01 |
| eGFR (mL/min/1.73 m²) | 202 | 77±20 | 104±10 | 75±8** | 48±9**, ## | <0.01 |
| Minimum-maximum | | 25–132 | 91–132 | 60–90 | 25–60 | |
| 25th percentile, median, 75th percentile | | 64, 76, 90 | 96, 103, 107 | 69, 76, 83 | 44, 50, 54 | |

eGFR, estimated glomerular filtration rate; LDL, low-density lipoprotein; HDL, high-density lipoprotein; RAAS, renin-angiotensin-aldosterone system; DPP-4, dipeptidyl peptidase-4; GLP-1, glucagon-like peptide-1; uPE, urinary protein excretion; uACR, urinary albumin-to-creatinine ratio; L-FABP, liver-type fatty acid-binding protein.

¶ N: number estimated.

† Diabetic retinopathy includes simple, preproliferative and proliferative retinopathy.

‡ RAAS inhibitors include angiotensin-converting enzyme inhibitors, angiotensin II receptor blockers and aldosterone receptor antagonists.

* *P*<0.05

** *P*<0.01 vs. corresponding value in the high eGFR group

# *P*<0.05

## *P*<0.01 vs. corresponding value in normal eGFR group.

significantly larger in the normal and low eGFR groups than in the high eGFR group. Although the changes in uACR and urinary L-FABP were larger in the low eGFR group than in the normal and high eGFR groups, there were no significant differences. In the all subjects, the changes in the uPE, uACR and urinary L-FABP showed significantly negative correlations with the corresponding values at baseline (**S1 Fig**). While the eGFR was significantly reduced from baseline at 1 to 12 months in both the high and normal eGFR groups after the initiation of luseogliflozin, the eGFR did not significantly differ in the low eGFR group. The change in the eGFR from baseline was significantly larger in the high eGFR group than in the normal and low eGFR group. In the all of the subjects, the change in the eGFR showed significantly negative correlations with the baseline eGFR (**S2 Fig**).

**Fig 2A** shows the changes in the eGFR in 145 subjects who continued visiting over 12 months before luseogliflozin administration. In the low eGFR group (*n* = 29), the eGFR did not change significantly after luseogliflozin administration, although the eGFR gradually decreased before the initiation of luseogliflozin. The change in the eGFR (2±6 mL/min/1.73 m²) was significantly (*P*<0.01) improved after luseogliflozin administration compared to before administration (-5±7 mL/min/1.73 m²). The eGFR significantly decreased after the initiation of luseogliflozin in the high (*n* = 35) eGFR group. The change in the eGFR was -0±12 mL/min/1.73 m² and -5±10 mL/min/1.73 m² before and after the initiation of luseogliflozin, respectively, in the high eGFR group. The change in the eGFR was -1±8 mL/min/1.73 m² and -0±10 mL/min/1.73 m² before and after the initiation of luseogliflozin, respectively, in the normal eGFR group (**S1 Table** and **Fig 2B**).

**Table 3** shows the relationships between the changes in the eGFR and uPE and the baseline clinical parameters in the FAS. The change in the eGFR showed a significant negative correlation with the systolic blood pressure, uACR and eGFR at baseline according to a multiple regression analysis with smoking history, diastolic blood pressure, hypo-HDL-cholesterolemia and HbA1c included as independent variables. The change in uPE was significantly associated with the uPE and uACR and at baseline according to a multiple regression analysis with gender, current drinker, diabetic retinopathy, cerebrovascular disease, HbA1c and serum creatinine concentration included as independent variables.

**Table 2. Changes in clinical parameters of the full analysis set.**

| | Baseline | 1 month | 3 months | 6 months | 9 months | 12 months | Change from baseline |
|---|---|---|---|---|---|---|---|
| Body weight (kg) | | | | | | | |
| All subjects ($n$ = 173) | 77.2±17.9 | 76.3±17.7** | 76.0±17.7** | 75.3±17.4** | 75.2±17.4** | 75.3±17.3** | -2.0±3.2 |
| High eGFR ($n$ = 44) | 77.5±20.6 | 76.5±20.2** | 75.6±19.9** | 75.4±19.8** | 75.4±19.5** | 75.5±19.3** | -2.0±3.5 |
| Normal eGFR ($n$ = 97) | 76.8±17.2 | 76.1±17.0** | 76.0±17.1** | 75.1±16.6** | 75.0±16.9** | 74.8±17.0**** | -1.9±3.3 |
| Low eGFR ($n$ = 32) | 77.9±16.7 | 76.8±16.3** | 76.4±16.8** | 75.8±16.7** | 75.6±16.3** | 75.9±16.0** | -2.0±2.3 |
| Systolic blood pressure (mmHg) | | | | | | | |
| All subjects ($n$ = 193) | 134±15 | 131±15** | 130±13** | 129±13** | 130±13** | 133±14** | -5±16 |
| High eGFR ($n$ = 47) | 137±16 | 133±16 | 132±12 | 131±12* | 130±14* | 131±13* | -4±16 |
| Normal eGFR ($n$ = 111) | 134±15 | 131±15* | 130±13** | 129±13** | 130±13* | 129±14* | -5±16 |
| Low eGFR ($n$ = 35) | 132±14 | 128±15* | 127±14** | 127±14* | 127±14* | 127±14* | -4±14 |
| Diastolic blood pressure (mmHg) | | | | | | | |
| All subjects ($n$ = 193) | 81±12 | 79±12** | 79±11** | 78±12** | 79±11* | 78±12* | -2±10 |
| High eGFR ($n$ = 47) | 84±14 | 81±13 | 83±12 | 81±12 | 79±13* | 81±14 | -2±13 |
| Normal eGFR ($n$ = 111) | 81±11 | 79±10 | 79±10** | 79±11* | 80±11 | 79±10 | -2±10 |
| Low eGFR ($n$ = 35) | 75±11##, $ $ | 73±12 | 73±13 | 73±12 | 74±11 | 72±12 | -3±9 |
| HbA1c (%) | | | | | | | |
| All subjects ($n$ = 201) | 8.3±1.4 | 7.9±1.2** | 7.7±1.1** | 7.5±1.0** | 7.4±0.9** | 7.5±1.0** | -0.8±1.2 |
| High eGFR ($n$ = 49) | 8.5±1.6 | 8.0±1.4** | 7.6±1.1** | 7.4±1.0** | 7.4±1.1** | 7.4±1.0** | -1.1±1.4 |
| Normal eGFR ($n$ = 116) | 8.3±1.4 | 7.9±1.2** | 7.7±1.1** | 7.5±0.9** | 7.4±0.9** | 7.5±1.1** | -0.8±1.1 |
| Low eGFR ($n$ = 36) | 8.1±1.1 | 8.0±1.1 | 7.8±1.0 | 7.8±1.0 | 7.7±0.9** | 7.7±0.9** | -0.4±0.8# |
| uPE (mg/dL) | | | | | | | |
| All subjects ($n$ = 196) | 22±47 | 14±31** | 14±38** | 11±29** | 9±16** | 9±17** | -13±41 |
| High eGFR ($n$ = 48) | 13±19 | 11±19* | 10±17** | 7±13** | 8±16** | 8±16** | -5±16 |
| Normal eGFR ($n$ = 112) | 23±50 | 14±37** | 15±47** | 14±37** | 8±15** | 9±17** | -14±42# |
| Low eGFR ($n$ = 36) | 33±58 | 17±20** | 16±27** | 10±16** | 13±17** | 11±19** | -22±55# |
| uACR (mg/gCr) | | | | | | | |
| Allsubjects ($n$ = 157) | 117±274 | | | 77±183** | | 73±175** | -45±143 |
| High eGFR ($n$ = 43) | 91±162 | | | 51±86** | | 56±99* | -35±105 |
| Normal eGFR ($n$ = 88) | 112±249 | | | 83±201** | | 77±193** | -35±100 |
| Low eGFR ($n$ = 26) | 178±449 | | | 99±234** | | 84±210** | -94±266 |
| Urinary L-FABP (µg/gCr) | | | | | | | |
| All subjects ($n$ = 50) | 5.2±6.9 | | | 3.7±6.6** | | 3.2±3.9** | -2.0±4.2 |
| High eGFR ($n$ = 14) | 3.6±2.0 | | | 2.3±1.4** | | 2.2±1.3** | -1.4±1.9 |
| Normal eGFR ($n$ = 22) | 3.6±3.1 | | | 2.2±1.9* | | 2.3±1.6* | -1.4±2.5 |
| Low eGFR ($n$ = 14) | 9.2±11.7 | | | 7.5±11.6 | | 5.7±6.4* | -3.5±7.0 |
| eGFR (mL/min/1.73 m$^2$) | | | | | | | |
| All subjects ($n$ = 202) | 77±20 | 75±20** | 75±20** | 75±20** | 75±20** | 75±20** | -2±10 |
| High eGFR ($\boldsymbol{n}$ = 49) | 104±10 | 99±13** | 99±12** | 98±13** | 100±14* | 99±12** | -5±10 |
| Normal eGFR ($n$ = 116) | 75±8## | 74±11* | 73±11** | 73±11** | 73±12** | 74±13* | -1±10# |
| Low eGFR ($n$ = 37) | 48±9##, $ $ | 48±10 | 48±11 | 48±11 | 50±10 | 49±12 | 1±7##, $ |

eGFR, estimated glomerular filtration rate; uPE, urinary protein excretion; uACR, urinary albumin-to-creatinine ratio; L-FABP, liver-type fatty acid-binding protein.

* $P<0.05$

** $P<0.01$ vs. corresponding value at baseline

# $P<0.05$

## $P<0.01$ vs. corresponding value in the high eGFR group

$ $P<0.05$

$ $ $P<0.01$ vs. corresponding value in the normal eGFR group.

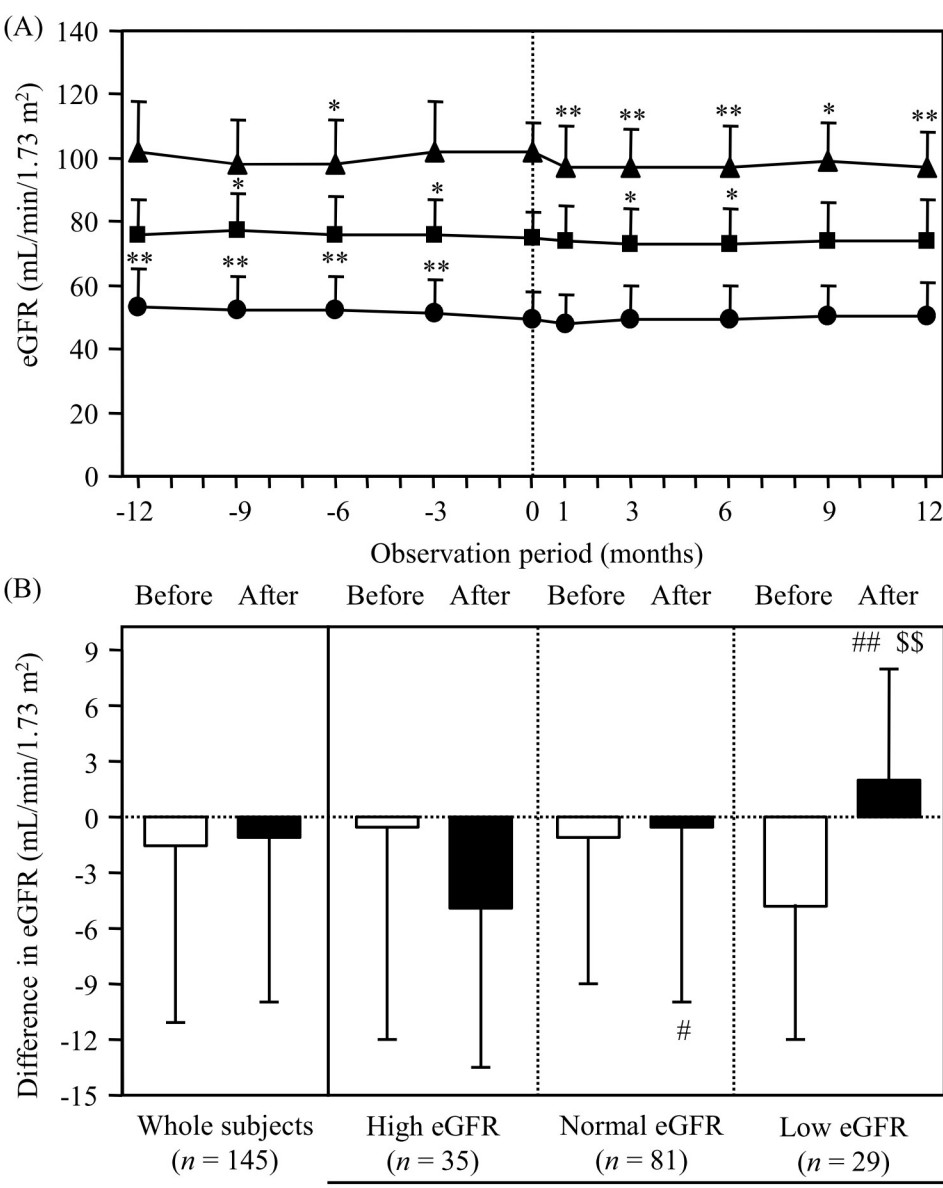

**Fig 2. Changes in the eGFR in the groups according to the eGFR at baseline (0 month) before and after the initiation of luseogliflozin ($n$ = 145).** (A) The closed triangles, squares and circles indicate subjects in the high ($\geq$90 mL/min/1.73 m$^2$, $n$ = 35), normal (60 to <90 mL/min/1.73 m$^2$, $n$ = 81) and low (60 mL/min/1.73 m$^2$, $n$ = 29) eGFR groups, respectively. $^*P$ <0.05 and $^{**}P$ <0.01 vs. baseline (0 months) value. (B) Open and closed bars indicate the differences in the eGFR during 12 months before and after the initiation of luseogliflozin, respectively. #$P$ <0.05 and ##$P$ <0.01 vs. corresponding value in the high eGFR group, \$ \$P <0.01 vs. corresponding value before the initiation. eGFR, estimated glomerular filtration rate.

## Safety

In the SAS, luseogliflozin was discontinued in 24 patients (11%) at the onset of AEs (**Fig 1**). AEs considered to be associated with luseogliflozin administration, including urogenital infection, increased urine volume, volume depletion, skin eruption and gastrointestinal symptoms, were observed within 100 days after the initiation of luseogliflozin administration (**S2 Table**).

**Table 3. Relationship between the changes in the eGFR and uPE and the clinical parameters at baseline in the full analysis set.**

| | Changes in the eGFR | | | | Changes in the uPE | | | |
|---|---|---|---|---|---|---|---|---|
| | single regression | | Multiple regression | | single regression | | Multiple regression | |
| | Regression coefficient | P | β | P | Regression coefficient | P | β | P |
| Male gender | -1.442 | 0.06 | | | -7.686 | 0.02 | 3.780 | 0.20 |
| Age (/years) | 0.057 | 0.36 | | | 0.122 | 0.65 | | |
| Duration of diabetes (/years) | 0.016 | 0.87 | | | 0.354 | 0.39 | | |
| Smoking history | -3.646 | 0.01 | -2.342 | 0.14 | -3.261 | 0.62 | | |
| Current drinker | -0.223 | 0.89 | | | -14.425 | 0.04 | -0.988 | 0.72 |
| Body weight (/kg) | -0.032 | 0.42 | | | -0.203 | 0.27 | | |
| Body mass index (/kg/m$^2$) | 0.022 | 0.86 | | | -0.343 | 0.56 | | |
| Systolic blood pressure (/mmHg) | -0.141 | <0.01 | -0.119 | 0.04 | -0.326 | 0.11 | | |
| Diastolic blood pressure (/mmHg) | -0.125 | 0.03 | 0.029 | 0.69 | -0.176 | 0.48 | | |
| Diabetic retinopathy† | -1.145 | 0.43 | | | -24.577 | <0.01 | 3.690 | 0.68 |
| Diabetic nephropathy | -3.807 | <0.01 | | | -25.153 | <0.01 | | |
| Diabetic peripheral neuropathy | 2.301 | 0.21 | | | -3.480 | 0.88 | | |
| Cerebrovascular disease | -1.873 | 0.52 | | | -46.259 | <0.01 | -2.393 | 0.68 |
| Coronary heart disease | -0.771 | 0.70 | | | 4.879 | 0.58 | | |
| Peripheral arterial disease | -0.585 | 0.91 | | | -3.480 | 0.88 | | |
| Obesity | 0.706 | 0.67 | | | -8.603 | 0.27 | | |
| Hypertension | 0.306 | 0.85 | | | -10.714 | 0.11 | | |
| Hyper-LDL-cholesterolemia | 0.886 | 0.62 | | | -11.115 | 0.14 | | |
| Hypo-HDL-cholesterolemia | -1.949 | 0.01 | -2.365 | 0.16 | -12.746 | 0.08 | | |
| Hyperuricemia | 1.940 | 0.29 | | | -4.850 | 0.53 | | |
| RAAS inhibitors use‡ | 2.549 | 0.06 | | | -7.634 | 0.19 | | |
| Calcium channel blockers use | -0.421 | 0.76 | | | -3.680 | 0.53 | | |
| Cholesterol lowering agents use | -0.942 | 0.55 | | | -11.608 | 0.08 | | |
| Urate lowering agents use | 0.381 | 0.85 | | | -10.011 | 0.24 | | |
| Anti-diabetic agents use | | | | | | | | |
| Metformin | -1.278 | 0.37 | | | 2.058 | 0.74 | | |
| Sulfonylureas | 2.486 | 0.15 | | | -7.260 | 0.33 | | |
| Thiazolidinediones | 2.413 | 0.24 | | | 7.381 | 0.40 | | |
| α-glucosidase inhibitors | -0.860 | 0.68 | | | 2.685 | 0.77 | | |
| Glinides | 4.038 | 0.20 | | | 3.414 | 0.80 | | |
| DPP-4 inhibitors | 2.413 | 0.09 | | | 1.598 | 0.80 | | |
| GLP-1 receptor agonists | -0.342 | 0.92 | | | -24.641 | 0.09 | | |
| Insulin | -0.509 | 0.74 | | | -7.031 | 0.28 | | |
| Number of anti-diabetic agents | 0.771 | 0.23 | | | -1.239 | 0.65 | | |
| uPE (/mg/dL) | -0.024 | 0.10 | | | -0.815 | <0.01 | -0.891 | <0.01 |
| uACR (/mg/gCr) | -0.007 | 0.01 | -0.006 | 0.02 | -0.065 | <0.01 | -0.019 | <0.01 |
| L-FABP (/μg/gCr) | 0.056 | 0.73 | | | -0.486 | 0.18 | | |
| HbA1c (/%) | -1.042 | 0.04 | -0.943 | 0.09 | -4.413 | 0.04 | -1.189 | 0.14 |
| LDL-cholesterol (/mmol/L) | 0.735 | 0.38 | | | 6.426 | 0.08 | | |
| HDL-cholesterol (/mmol/L) | 2.687 | 0.20 | | | 11.779 | 0.22 | | |
| AST (/IU/L) | -0.005 | 0.88 | | | -0.072 | 0.61 | | |
| ALT (/IU/L) | 0.006 | 0.79 | | | 0.273 | 0.78 | | |
| γGTP (/IU/L) | 0.015 | 0.20 | | | 0.003 | 0.96 | | |
| Creatinine (/μmol/L) | 0.045 | 0.14 | | | -0.267 | 0.04 | -7.274 | 0.15 |
| Uric acid (/μmol/L) | 0.014 | 0.20 | | | -0.072 | 0.17 | | |

(*Continued*)

**Table 3.** (Continued)

| | Changes in the eGFR | | | | Changes in the uPE | | | |
| | single regression | | Multiple regression | | single regression | | Multiple regression | |
| | Regression coefficient | *P* | β | *P* | Regression coefficient | *P* | β | *P* |
|---|---|---|---|---|---|---|---|---|
| eGFR (/mL/min/1.73 m$^2$) | -0.115 | <0.01 | -0.076 | 0.04 | 0.185 | 0.20 | | |

eGFR, estimated glomerular filtration rate; uPE, urinary protein excretion; LDL, low-density lipoprotein; HDL, high-density lipoprotein; RAAS, renin-angiotensin-aldosterone system; DPP-4, dipeptidyl peptidase-4; GLP-1, glucagon-like peptide-1; uACR, urinary albumin-to-creatinine ratio; L-FABP, liver-type fatty acid-binding protein.

† Diabetic retinopathy includes simple, preproliferative and proliferative retinopathy.

‡ RAAS inhibitors include angiotensin-converting enzyme inhibitors, angiotensin II receptor blockers and aldosterone receptor antagonists.

The frequencies of AEs recorded during the observation period among the eGFR groups in the SAS are shown in S3 Table. Overall AEs were recorded in 86 cases among 238 patients (36%) and were significantly more frequent in the low eGFR groups (66%) than in the high (30%) and normal (28%) eGFR groups. Although the frequencies of volume depletion (11%) and skin itching/eruption (9%) were higher in the low eGFR group than in the other groups, the difference was not statistically significant among the three groups.

## Discussion

The eGFR was preserved in the low eGFR group after luseogliflozin administration but gradually decreased before the administration in the present study. Several large-scale clinical trials have demonstrated the renoprotective effects, including the improvement of proteinuria and preservation of the eGFR, by SGLT2 inhibitors [2–9]. Although some had studies investigated the efficacy of SGLT2 inhibitors, including luseogliflozin, in a small number of Japanese patients with type 2 diabetes and renal impairment, none had compared the results between subjects with a normal renal function and those with an impaired function [40–42]. The eGFR showed a transient decline after the initiation of SGLT2 inhibitor treatment and subsequent normalization over time in previous clinical studies [2, 9]. In the present study, the eGFR showed similar fluctuations after the initiation of luseogliflozin in the high and normal eGFR groups.

One highlight of the current study is that the renoprotective effect of luseogliflozin was demonstrated in the patients with a low eGFR who had shown a declining eGFR before the luseogliflozin therapy. Blood pressure control using antihypertensive agents, such as renin-angiotensin-aldosterone system (RAAS) inhibitors, is widely known to be useful for preventing the progression of renal impairment in patients with type 2 diabetes [43, 44]. Urate-lowering agents also ameliorate the decline in the kidney function in patients with hyperuricemia and chronic kidney disease, including diabetic kidney disease (DKD) [45]. The observation of a renoprotective effect by luseogliflozin in patients with type 2 diabetes who were already being frequently treated with RAAS inhibitors and/or urate-lowering agents in a real-world clinical setting was considered valuable.

Albuminuria and a low eGFR are established risk factors for end-stage kidney disease and cardiovascular events in patients with type 2 diabetes, and reducing the uACR and preserving the eGFR are helpful for suppressing the development of renal and cardiovascular events [25–30, 43, 44]. Although the blood glucose-lowering effect was inferior in the low eGFR group compared to the effect in subjects with a preserved eGFR, the blood pressure and body weight decreased similarly in all three eGFR groups, regardless of the degree of renal impairment, in the present study as in previous reports [31]. The uPE, uACR and urinary L-FABP were all

inversely correlated with the corresponding values at baseline, and as a result, the eGFR improved in the low eGFR group. These results do not contradict our findings report in non-elderly and elderly patients with type 2 diabetes treated with empagliflozin, whose eGFR values at the baseline were 84.5 mL/min/1.73 m$^2$ and 67.2 mL/min/1.73 m$^2$, respectively [24]. The reductions in the body weight, blood pressure and HbA1c caused by SGLT2 inhibitors seem to be associated with urinary glucose excretion. However, why the subjects in the low eGFR group in the present study, whose improvement in HbA1c was inferior, showed similar changes in the body weight and blood pressure to the normal and high eGFR groups is unclear.

The urinary L-FABP as well as uPE and uACR values were reduced by luseogliflozin in the present study, regardless of the eGFR at the baseline. While albuminuria reflects the glomerular injury in the kidneys, urinary L-FABP from proximal tubules is increased under conditions in which fatty acids are loaded into the proximal tubules, such as conditions of ischemia and exposure to nephrotoxic substances [37–39]. The interstitium, including the renal tubules but not the glomeruli, anatomically occupies most of the kidney, and tubulointerstitial injury is known to be more closely related to the renal prognosis than to glomerular lesions in patients with chronic kidney disease [46]. Because urinary L-FABP is a predictive biomarker for the renal prognosis and incidence of cardiovascular diseases in patients with type 2 diabetes, the combination of measurements of uACR and urinary L-FABP is considered useful in the prevention of diabetic angiopathies [37–39]. Although whether or not a reduction in urinary L-FABP is commonly observed in patients with type 2 diabetes treated with SGLT2 inhibitors is unclear, patients showing a reduction in both the uACR and urinary L-FABP may have a better prognosis with regard to DKD than those without such reductions. The association of the changes in L-FABP after luseogliflozin administration with the incidence of renal and cardiovascular events should be investigated in a future study.

Although total AEs were significantly more common in the low eGFR group in the present study than in the other groups, there were no significant differences in the frequency of AEs that were specific for SGLT2 inhibitors. This result is likely due to the phenomenon wherein non-specific AEs are generally observed more frequently in the elderly and/or patients with renal impairment than in others [15, 24]. Samukawa *et al.* also reported that the frequency of AEs specific to luseogliflozin did not increase in a small number of patients with type 2 diabetes and a low eGFR [33]. Pharmacokinetic parameters, such as the area under the concentration-time curve (AUC) from 0 to infinity after the administration of luseogliflozin, are similar regardless of the degree of renal impairment in patients with type 2 diabetes [31], although the AUCs of other SGLT2 inhibitors are increased in patients with renal impairment [47, 48]. The fact that an increased risk of AEs related to increased drug exposure in patients with renal impairment is not an issue with luseogliflozin seems advantageous. However, it should be emphasized that there were many adverse events in the low eGFR group, even if they were not specific to luseogliflozin. Furthermore, adverse events related to the renal prognosis, such as uremia and kidney injury were observed in the subjects of low eGFR group who were excluded from the FAS. Thus, careful interpretation is required in relation to the renoprotective effect of luseogliflozin in the low eGFR group.

The present study population showed a male predominance. In 2017, the prevalence of diabetes in men (18.1%) is greater than that in women (10.5%) in the Japanese adult population [10]. However, the proportion of men in this study seems to be higher in comparison to our other studies (58%), which were conducted around the same time [30, 39]. Although no previous reports have investigated sex difference in the prevalence of obesity in Japanese diabetic patients, obesity is predominant in all generations of the Japanese general population [10]. Because it was likely that obese subjects were selected at a higher rate for treatment using

SGLT2 inhibitors, the sex difference in the prevalence of obesity is considered to have caused the sex bias in the subjects of the present study. Several limitations associated with the present study warrant mention. First, the present study was unable to investigate the adherence to non-pharmacological therapy, such as salt restriction, and pharmacological therapy. Because medication adherence is poor in younger patients with type 2 diabetes [49], the patients' age in the eGFR groups may have affected the outcome of the present study. Second, the eGFR was calculated by the formula recommended by the Japanese Society of Nephrology [34] and not measured using inulin clearance, which is the gold standard for determining the GFR. Because the eGFR was calculated using the serum creatinine concentration, it should be noted that the kidney function may have been overestimated in elderly individuals with a reduced skeletal muscle mass. Luseogliflozin administration, similarly to other SGLT2 inhibitors, reportedly reduces the skeletal muscle mass in patients with type 2 diabetes, although the reduction is less than the body fat reduction [50]. Because a decrease in a serum creatinine concentration caused by a reduced skeletal muscle mass after the initiation of luseogliflozin may affect the preservation of the eGFR in elderly patients with renal dysfunction, the body composition should have been examined before and after the study.

However, even with these limitations, we believe that luseogliflozin is effective for protecting the kidney function and is clinically safe to administer to patients with type 2 diabetes and renal impairment. The observation of a renoprotective effect of luseogliflozin in patients with type 2 diabetes without an increase in AEs specific to SGLT2 inhibitors is considered valuable as real-world data.

## Conclusion

Luseogliflozin works to the preserve renal function in the medium term in patients with type 2 diabetes and renal impairment without an increase in specific AEs.

## Supporting information

**S1 Dataset. Dataset in the present study.**
(XLSB)

**S1 Fig. Relationships between the changes in uPE, uACR and urinary L-FABP and the corresponding values at the baseline.** uPE, urinary protein excretion; uACR, urinary albumin-to-creatinine ratio; L-FABP, liver-type fatty acid-binding protein.
(TIF)

**S2 Fig. Relationship between the change in the eGFR and the eGFR at the baseline.** eGFR, estimated glomerular filtration rate.
(TIF)

**S1 Table. Changes in the eGFR in 145 patients who had been treated over 12 months before the initiation of luseogliflozin administration.**
(DOCX)

**S2 Table. Adverse events that caused discontinuation of luseogliflozin in the safety analysis set.**
(DOCX)

**S3 Table. Adverse events during the observation period in the safety analysis set.**
(DOCX)

## Acknowledgments

The authors thank Tomoko Koyanagi in the secretarial section of Edogawa Hospital for her valuable help with data collection.

## Author Contributions

**Conceptualization:** Hiroyuki Ito.

**Data curation:** Hiroyuki Ito.

**Formal analysis:** Hiroyuki Ito.

**Funding acquisition:** Hiroyuki Ito.

**Investigation:** Suzuko Matsumoto, Takuma Izutsu, Eiji Kusano, Jiro Kondo, Hideyuki Inoue, Shinichi Antoku, Tomoko Yamasaki, Toshiko Mori, Michiko Togane.

**Visualization:** Hiroyuki Ito.

**Writing – original draft:** Hiroyuki Ito.

**Writing – review & editing:** Suzuko Matsumoto, Takuma Izutsu, Eiji Kusano, Jiro Kondo, Hideyuki Inoue, Shinichi Antoku, Tomoko Yamasaki, Toshiko Mori, Michiko Togane.

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
