## [Decision Letter · Decision Letter 0]

10 Feb 2021

PONE-D-21-00301

Different renoprotective effects of luseogliflozin depend on the renal function at the baseline in patients with type 2 diabetes: A retrospective study during 12 months before and after initiation

PLOS ONE

Dear Dr. Ito,

Thank you for submitting your manuscript to PLOS ONE. After careful consideration, we feel that it has merit but does not fully meet PLOS ONE’s publication criteria as it currently stands. Therefore, we invite you to submit a revised version of the manuscript that addresses the points raised during the review process.Please incorporate  some of the revisions as  suggested by the  reviewers.

We look forward to receiving your revised manuscript.

Kind regards,

Bhagwan Dass, MD

Academic Editor

PLOS ONE

Journal Requirements:

2. We note that your retrospective study was approved in 2018 but that data was collected for patients in 2019. Please clarify whether the ethical approval included permission to keep gathering retrospective data after approval.

"This work was partly supported by Taisho Pharmaceutical Co., Ltd (Tokyo, Japan)."

"H Ito has received lecture fees from Taisho Pharmaceutical Co., Ltd., Eli Lilly Japan KK, Boehringer Ingelheim, Takeda Pharmaceutical Company Ltd., Sanofi KK, Novo Nordisk Pharma Ltd., MSD KK, Novartis Pharma KK, Astellas Pharma, Daiichi Sankyo Company, Terumo Corporation, Mochida Pharmaceuticals, Teijin Pharma, Kissei Pharmaceuticals, Kowa Pharmaceuticals, Mitsubishi Tanabe Pharma Corporation, Sanwa Kagaku Kenkyusho, Dainippon Sumitomo Pharma, AstraZeneca KK, Kyowa Hakko Kirin, Shionogi and Co, Otsuka Pharmaceutical Co., Ltd., Bayer Yakuhin, Ltd., and Santen Pharmaceutical Co., Ltd., and has received consulting fee from Becton, Dickinson and Company. S Matsumoto has received lecture fees from Novo Nordisk Pharma Ltd., Astellas Pharma, and AstraZeneca KK. T Izutsu has received lecture fees from Sanofi KK, Taisho Pharmaceutical Co., Ltd., Kyowa Hakko Kirin, Bayer Yakuhin, Ltd., and Mitsubishi Tanabe Pharma Corporation. S Antoku has received lecture fees from Kyowa Hakko Kirin, Sanofi KK, Kyowa Hakko Kirin, Taisho Pharmaceutical Co., Ltd., Daiichi Sankyo Company, and Otsuka Pharmaceutical Co., Ltd. E Kusano, J Kondo, H Inoue, T Yamasaki, T Mori and M Togane have no conflict of interest."

We note that you received funding from a commercial source: Taisho Pharmaceutical Co., Ltd.

Reviewers' comments:

Reviewer's Responses to Questions

**Comments to the Author**

1. Is the manuscript technically sound, and do the data support the conclusions?

Reviewer #1: Yes

Reviewer #2: Yes

2. Has the statistical analysis been performed appropriately and rigorously? 

Reviewer #1: Yes

Reviewer #2: I Don't Know

3. Have the authors made all data underlying the findings in their manuscript fully available?

Reviewer #1: Yes

Reviewer #2: Yes

4. Is the manuscript presented in an intelligible fashion and written in standard English?

Reviewer #1: No

Reviewer #2: Yes

5. Review Comments to the Author

Reviewer #1: This is a well done safety study for luseogliflozin in patients with diabetes and renal disease.

The major strengths of the study included a follow up of 12 months in this patient sub population with renal disease. Also, elderly patients with long-standing diabetes were studied for safety issues which is an important factor in real-world use of these medications.

Analysis was done in three sub groups based on high, normal and low GFR and luseogliflozin use was found to be renoprotective in the low eGFR group.

One of the limitations was the removal of 36 patients from the study based on adverse events occurring in the first 100 days. These patients could have been included in an Intention-to-treat analysis. Also, the adverse events in this group were renal related as well which could have affected overall study data.

2. The study population was skewed towards men, with 71% of the study population being men. Is it related to baseline demographics in the Japanese population?

3. More adverse effects occurred in the low eGFR group, which is important, even though the authors state that these were not necessarily SGLT-2 related

4. In most studies of SGLT-2 inhibitors, there is a transient decline in GFR, and this normalizes over time. What is your hypothesis for the persistent decline in the high and normal eGFR groups?

5. There are grammatical and typographical errors that will need to be fixed. Minor changes.

Reviewer #2: Overall well written manuscript. SGLT-2 inhibitors have become an important armamentarium in Type 2 Diabetes Mellitus and it's indication will become broader as it's use in heart failure without diabetes increase in next few years. The beneficial effect of SGLT-2 inhibitors with respect to kidney has been put to rest with many landmark trials like CREDENCE (NEJM 2019). Baseline eGFR was ~ 56.3. Similar results has been shown in trials CANVAS and DECLARE-TIMI. I wonder if this study adds to the information that we don't know already.

Few other things:

Line 83- add "with" in front of diabetes

Line 215- Replace "Whole" with "All" and in other parts of manuscript if other reviewers and editors also agree

6. PLOS authors have the option to publish the peer review history of their article (what does this mean?). If published, this will include your full peer review and any attached files.

Reviewer #1: No

Reviewer #2: No

---

## [Author Response · Author response to Decision Letter 0]

25 Feb 2021

Response to the academic editor

Response: We amended the manuscript to meet PLOS ONE’s style requirements.

2. We note that your retrospective study was approved in 2018 but that data was collected for patients in 2019. Please clarify whether the ethical approval included permission to keep gathering retrospective data after approval.

Response: We apologize for the mistake in the year. We amended the description (2019 -> 2018) in the Study design and patients and Figure 1 in the revised manuscript.

"This work was partly supported by Taisho Pharmaceutical Co., Ltd (Tokyo, Japan)."

Response: We amended the Funding Statement in the revised manuscript and added the Funding Statement within our cover letter.

Response: We amended the Funding Information although the grant number was not issued.

"H Ito has received lecture fees from Taisho Pharmaceutical Co., Ltd., Eli Lilly Japan KK, Boehringer Ingelheim, Takeda Pharmaceutical Company Ltd., Sanofi KK, Novo Nordisk Pharma Ltd., MSD KK, Novartis Pharma KK, Astellas Pharma, Daiichi Sankyo Company, Terumo Corporation, Mochida Pharmaceuticals, Teijin Pharma, Kissei Pharmaceuticals, Kowa Pharmaceuticals, Mitsubishi Tanabe Pharma Corporation, Sanwa Kagaku Kenkyusho, Dainippon Sumitomo Pharma, AstraZeneca KK, Kyowa Hakko Kirin, Shionogi and Co, Otsuka Pharmaceutical Co., Ltd., Bayer Yakuhin, Ltd., and Santen Pharmaceutical Co., Ltd., and has received consulting fee from Becton, Dickinson and Company. S Matsumoto has received lecture fees from Novo Nordisk Pharma Ltd., Astellas Pharma, and AstraZeneca KK. T Izutsu has received lecture fees from Sanofi KK, Taisho Pharmaceutical Co., Ltd., Kyowa Hakko Kirin, Bayer Yakuhin, Ltd., and Mitsubishi Tanabe Pharma Corporation. S Antoku has received lecture fees from Kyowa Hakko Kirin, Sanofi KK, Kyowa Hakko Kirin, Taisho Pharmaceutical Co., Ltd., Daiichi Sankyo Company, and Otsuka Pharmaceutical Co., Ltd. E Kusano, J Kondo, H Inoue, T Yamasaki, T Mori and M Togane have no conflict of interest."

We note that you received funding from a commercial source: Taisho Pharmaceutical Co., Ltd.

Response: We amended the Competent interest Statement in the revised manuscript and added the Competent interest Statement within our cover letter.

 

Response to the reviewer #1

1. One of the limitations was the removal of 36 patients from the study based on adverse events occurring in the first 100 days. These patients could have been included in an Intention-to-treat analysis. Also, the adverse events in this group were renal related as well which could have affected overall study data.

Response: Because we cannot analyze the clinical data during 12 months in the subjects whose luseogliflozin administration was stopped, groups according to the baseline eGFR were added to S5 Table of the revised manuscript. Adverse events related to the renal prognosis, such as uremia and kidney injury were observed in the low eGFR group. Thus, we added an explanation to the Discussion section of the revised manuscript indicating that careful interpretation is required in relation to the renoprotective effect of luseogliflozin in the low eGFR group when considering subjects, including the SAS. 

2. The study population was skewed towards men, with 71% of the study population being men. Is it related to baseline demographics in the Japanese population?

Response: In 2017, the prevalence of diabetes in the Japanese adult population was greater in men (men, 18.1%; women, 10.5%) (reference 10). However, the proportion of men in this study seems to be higher in comparison to our other studies, which were conducted around the same time (58% in reference 30,39) as the reviewer pointed out. Although there are no previous reports investigating sex differences in the prevalence of obesity in Japanese diabetic patients, obesity is predominant in all generations of the Japanese general population (reference 10). Because it is highly likely that obese subjects were selected for treatment using SGLT2 inhibitors, the difference in the prevalence of obesity between men and women appears to have caused a sex difference in the subjects of the present study

We added an explanation about these points to the Discussion section of the revised manuscript.

3. More adverse effects occurred in the low eGFR group, which is important, even though the authors state that these were not necessarily SGLT-2 related

Response: In the Discussion section of the revised manuscript, we added a passage noting that caution is required to detect adverse events in the low eGFR group, as the reviewer suggested.

4. In most studies of SGLT-2 inhibitors, there is a transient decline in GFR, and this normalizes over time. What is your hypothesis for the persistent decline in the high and normal eGFR groups?

Response: In this study, the eGFR showed a transient decline after the start of luseogliflozin, and did not decrease thereafter in the high eGFR group or recovered to the baseline value in the normal eGFR group (S4 Table). We added a passage indicating that these changes did not contradict the previous studies to the Discussion section of the revised manuscript.

5. There are grammatical and typographical errors that will need to be fixed. Minor changes.

Response: The manuscript was revised after editing by a professional editor who is a native speaker of English.

 

Response to the reviewer #2

Line 83- add "with" in front of diabetes

Response: We added “with” in front of diabetes in the revised manuscript.

Line 215- Replace "Whole" with "All" and in other parts of manuscript if other reviewers and editors also agree

Response: We replaced “whole” to “all” in the revised manuscript.

---

## [Editor Report · Decision Letter 1]

2 Mar 2021

Different renoprotective effects of luseogliflozin depend on the renal function at the baseline in patients with type 2 diabetes: A retrospective study during 12 months before and after initiation

PONE-D-21-00301R1

Dear Dr. Ito,

We’re pleased to inform you that your manuscript has been judged scientifically suitable for publication and will be formally accepted for publication once it meets all outstanding technical requirements.

Kind regards,

Bhagwan Dass, MD

Academic Editor

PLOS ONE
---

## [Editor Report · Acceptance letter]

5 Mar 2021

PONE-D-21-00301R1 

Different renoprotective effects of luseogliflozin depend on the renal function at the baseline in patients with type 2 diabetes: A retrospective study during 12 months before and after initiation 

Dear Dr. Ito:

I'm pleased to inform you that your manuscript has been deemed suitable for publication in PLOS ONE. Congratulations! Your manuscript is now with our production department. 

Kind regards, 

on behalf of

Dr. Bhagwan Dass 

Academic Editor

PLOS ONE